# An Insight into the Arising Role of MicroRNAs in Hepatocellular Carcinoma: Future Diagnostic and Therapeutic Approaches

**DOI:** 10.3390/ijms24087168

**Published:** 2023-04-12

**Authors:** Evangelos Koustas, Eleni-Myrto Trifylli, Panagiotis Sarantis, Nikolaos Papadopoulos, Konstantinos Papanikolopoulos, Georgios Aloizos, Christos Damaskos, Nikolaos Garmpis, Anna Garmpi, Dimitris Matthaios, Michalis V. Karamouzis

**Affiliations:** 1Department of Biological Chemistry, Medical School, National and Kapodistrian University of Athens, 75, M. Asias Street, 11527 Athens, Greece; 2First Department of Internal Medicine, 417 Army Equity Fund Hospital, 11521 Athens, Greece; 3Second Department of Internal Medicine, 401 General Army Hospital of Athens, 11525 Athens, Greece; 4‘N.S. Christeas’ Laboratory of Experimental Surgery and Surgical Research, Medical School, National and Kapodistrian University of Athens, 11527 Athens, Greece; 5Renal Transplantation Unit, ‘Laiko’ General Hospital, 11527 Athens, Greece; 6Second Department of Propaedeutic Surgery, ‘Laiko’ General Hospital, Medical School, National and Kapodistrian University of Athens, 11527 Athens, Greece; 7First Department of Pathology, Medical School, National and Kapodistrian University of Athens, 11527 Athens, Greece; 8Oncology Department, General Hospital of Rhodes, 85100 Rhodes, Greece

**Keywords:** autophagy, hepatocellular carcinoma, drug resistance, microRNAs

## Abstract

Hepatocellular carcinoma (HCC) constitutes a frequent highly malignant form of primary liver cancer and is the third cause of death attributable to malignancy. Despite the improvement in the therapeutic strategies with the exploration of novel pharmacological agents, the survival rate for HCC is still low. Shedding light on the multiplex genetic and epigenetic background of HCC, such as on the emerging role of microRNAs, is considered quite promising for the diagnosis and the prediction of this malignancy, as well as for combatting drug resistance. MicroRNAs (miRNAs) constitute small noncoding RNA sequences, which play a key role in the regulation of several signaling and metabolic pathways, as well as of pivotal cellular functions such as autophagy, apoptosis, and cell proliferation. It is also demonstrated that miRNAs are significantly implicated in carcinogenesis, either acting as tumor suppressors or oncomiRs, while aberrations in their expression levels are closely associated with tumor growth and progression, as well as with local invasion and metastatic dissemination. The arising role of miRNAs in HCC is in the spotlight of the current scientific research, aiming at the development of novel therapeutic perspectives. In this review, we will shed light on the emerging role of miRNAs in HCC.

## 1. Introduction

Hepatocellular carcinoma (HCC) is considered one of the most frequently diagnosed malignancies on a global level. Based on recent epidemiological data, it is considered the third cause of death related to cancer, accounting for 830,000 deaths annually [1]. Despite the novel therapeutic approaches, HCC’s five-year survival rates still remain considerably low (≈18%) [2]. Meanwhile, there is a sex disparity in HCC, with increased incidence rates in males in comparison to females, which are mainly attributed to the molecular background of HCC [3]. More particularly, females express a liver-associated autosomal gene, the so-called *CYP39A1*, which plays a key role in the suppression of hepatic carcinogenesis and the development of HCC [3].

Starting by mentioning the most common risk factors of HCC development, such as viral hepatitis B and C and alcohol abuse, it has to be underlined that HCC is attributed ever more to non-viral factors such as non-alcoholic fatty liver disease [4].

Despite the novel therapeutic strategies such as immunotherapy, radiofrequency combined with immunomodulation and other local ablative approaches, as well as transplantation and novel techniques in the surgical management of this malignancy, survival is still relatively low. However, a better understanding of the molecular basis of HCC is considered pivotal for the optimal management of this malignancy [5,6]. The role of miRNAs is in the spotlight of the current scientific research for several malignancies, including HCC.

MicroRNAs (miRNAs) are short non-coding RNA sequences, composed of 19–25 nucleotides with a pivotal regulatory role for the cellular functional state, including autophagy, apoptosis, proliferation, and apoptosis, as well as metabolic pathways and signaling [7,8]. They are considered a double-edged sword for carcinogenesis as they play a binary role either as tumor suppressors or promoters, given their oncogenic behavior [9].

Aberrations in the expression levels of these regulatory molecules are closely implicated in tumorigenesis, as well as in tumor progression and dissemination, due to the disruption of fundamental cell functions [10].

In this review, we will shed light on the emerging role of miRNAs in hepatocellular cancer, as well as their manipulation for future therapeutic and diagnostic perspectives.

## 2. A Glance through the Canonical MiRNA Biogenesis

MicroRNAs (miRNAs) play a fundamental role in the regulation of genetic information, several signaling metabolic pathways, hematopoiesis, as well as programmed cell death and autophagy, while it is demonstrated that miRNAs are closely associated even with the microbiome [11].

The mechanism of miRNAs biogenesis is closely related to the DNA transcription of specific miRNAs coding genes, under the enzymatic action of RNA polymerase II. This procedure takes place inside the nucleus, by which primary miRNA (pri-miRNA) arises [12]. Then, the aforementioned miRNA molecule is cleaved under the action of the ribonuclease complex, the so-called DiGeorge Syndrome Critical Region 8 (DGCR8)–Drosha (ribonuclease III) [13]. The latter procedure gives rise to another miRNA molecule, the precursor miRNA (pre-miRNA), which is shorter than the pri-miRNA (≥1000 nucleotides), has a sequence of up to one hundred nucleotides [14].

Afterwards, the pre-miRNA is relocated to the cytoplasm via the action of Exportin 5 and RAS-related nuclear protein–guanosine-5′-triphosphate (Ran-GTP), where it is further cleaved by the Dicer-TRBP complex, with the former being an RNase III endonuclease that removes its terminal loop, and the latter an RNA-binding protein [15].

After the removal of the pre-miRNA terminal loop, the miRNA is presented as a short Duplex strand, which is further loaded into an RNA-induced silencing complex (RISC) complex. This complex is composed of Argonaute RISC Catalytic Component 2 (Ago2) that separates one of the strands, based on its target specificity [16]. More particularly, the so-called leading strand (the mature miRNA strand) is merged with the RISC complex, whereas the other is called the passenger strand and is further degraded [17,18]. The miRNA-RISC (miRISC) targets a partially complementary mRNA strand, binding on its 3′untranslated region (3′UTR), while the miRNA sequence that is complementary is called the seed sequence [14,19,20].

Furthermore, the attachment of the miRNA strand to the partially complementary mRNA sequence leads to the repression of the mRNA translation, as well as its silencing or degradation [21].

Last but not least, there is another pathway of biogenesis, the so-called non-canonical, which does not depend either on the action of the Drosha–DGCR8 or –DICER complexes [22], and there are also different seed regions from the canonical. We demonstrate a schematic representation of canonical miRNA biogenesis in Figure 1.

## 3. The Emerging Role of MiRNAs in HCC

There are several aberrations occurring in malignant cells, including modifications in the interactions between miRNAs and their targets, while modifications in their expression levels are closely implicated in tumor progression and dissemination, due to the disruption of fundamental cell functions [23]. It is widely demonstrated that the miRNA expression profiling of malignant tumors, the so-called miRNA signature, is closely related to the stage of the tumor, the tumor growth and progression, as well as with diagnosis, prognosis, and anti-neoplastic treatment response [24].

Several studies demonstrate the emerging role of miRNAs in HCC, being closely associated with tumor proliferative and invasive behavior, metastatic dissemination, as well as with the sensitivity of cancer cells to treatment [25].

### 3.1. The Oncogenic MiRNAs in HCC

There are several aberrations in the expression levels of miRNA that promote HCC progression. Some of the miRNAs are upregulated and facilitate tumor proliferation, progression, neoangiogenesis, as well as local invasion distant dissemination and drug resistance [26]. In this section, we will discuss the role of several miRNAs that act as tumor promoters, the so-called oncomiRs [27]. Some miRNAs are found to be upregulated or downregulated and they are closely associated with the proliferation of malignant hepatocytes [28].

More particularly, the miR-92a level is increased in HCC tissue and is closely associated with tumor proliferation and invasion via targeting FOXA2 [29], while miR-203a-3p.1 promotes metastatic dissemination via targeting IL24, which suppresses HCC metastasis [30]. MiR-4417 is another oncogenic miRNA that inhibits apoptosis and promotes tumor proliferation in HCC via its regulatory effect on Pyruvate Kinase Muscle 2 (PKM2) Phosphorylation and Tripartite Motif-Containing 35 (TRIM35) [31]. Additionally, miR-21 is usually found to be elevated in HCC, especially in NAFLD-associated cases [32]. This miRNA targets several pathways, including STAT3 and PI3K/AKT pathways, as well as TGF-β/SMADs and the cell cycle [32,33]. The implication of miR-21 in the cell cycle of hepatocytes is reflected by its inhibitory effect on hepatocellular carcinoma, down-regulated 1 (HEPN1), which otherwise suppresses the proliferation of HepG2 cells [33].

Meanwhile, miR-18a constitutes an oncogenic miRNA that enhances HCC cell proliferation and tumor migratory behavior via targeting Bcl2l10 [34], whereas the upregulation of miR-155-5p and miR-331-3p is correlated with HCC progression via downregulating the PI3K/Akt pathway through PTEN inhibition and E2F1 suppression, respectively [35,36].

Likewise, the upregulation of miR-765 and miR-302d causes HCC proliferation and migration via suppressing INPP4B and TGFBR2 expression, respectively [37,38]. Migration and metastatic dissemination are also correlated with an upregulated miR-487a level via interacting with SPRED2 and PIK3R1 [39], while an increased miR-454 level exerts a significant oncogenic effect on tumor-initiating cells (T-ICs), leading to recurrent HCC, metastatic dissemination, and resistance to therapeutic modalities [40].

Moreover, miR-346 induces tumor growth and progression by targeting breast cancer metastasis suppressor 1 (BRMS1) [41], while miR-3910 also promotes progression and migration of the disease via targeting MST1 and inducing the Hippo-YAP signaling pathway [42].

On top of that, miR-873 induces HCC growth and metastatic dissemination via the Warburg effect, which is mediated by the AKT/mTOR pathway [43]. A list of several oncogenic miRNAs and their implication in HCC is demonstrated in Table 1.

### 3.2. MiRNAs as Tumor Suppressors for HCC

Recent studies have revealed that the loss or the decrease in tumor-suppressive miRNA expression levels could potentially lead to oncogenesis. More particularly, miR-199a-5p expression is closely associated with glycometabolism in hepatocytes, while in the case of hepatocarcinogenesis, it reprograms the glycolysis of the malignant cells via binding to 3′ UTR hexokinase 2(HK2) mRNA [44]. In addition, it has to be underlined that tumor-suppressing miR-199a is one of the most downregulated miRNAs in HCC specimens, while the enhancement of its expression levels could be notably beneficial as it can induce the impairment of tumor growth via interfering with aerobic glycolysis, which is crucial for their proliferation [45]. However, it has recently been shown that miR-199a/b-3p has also a tumor-inhibiting effect via interfering with the MAPK/ERK pathway by suppressing the p21-activated kinase 4 (PAK4) [46].

Moreover, miRNA-148-3p targets SMAD2, resulting in the downregulation of the latter and tumor suppression [47], while miR-195 also suppresses HCC development via interacting with the G1-S cell cycle regulatory genes such as CDC25A, CDK6, CDK4, as well as CCNE1. Based on the aforementioned, miR-195 can significantly alter the cell cycle of malignant hepatocytes [48].

Furthermore, miR-296-5p is another tumor-suppressing miRNA for HCC tissues which suppresses the HCC stem cell lines and the Neuregulin-ERBB signaling pathway, as well as limits the EMT phenomenon [49,50]. Additionally, miR-206 constitutes a pivotal suppressing miRNA, as it could totally inhibit HCC development in mice, when it is delivered in the malignant hepatic tissue [51]. The above phenomenon is attributed to the significant targets of this miRNA, such as the cyclin-dependent kinase (CDK) enzymes CDK6 and CCND1 that regulate the progression of the cell cycle and C-MET signaling pathway via targeting cMET protein [51].

Meanwhile, it has been demonstrated that miR-29a suppresses metastatic dissemination and invasion via targeting VEGFA, LOXL2, and LOX [52]. However, the expression of miR-29a is lowered in HCC tissue, and its enhancement could be potentially used as a therapeutic strategy [53]. Additionally, miR-766-3p could be also characterized as tumor-suppressive miRNA in HCC as it is closely associated with the Wnt/β-catenin signaling pathway, which constitutes a pivotal pathway for the physiological differentiation and multiplication of hepatocytes [54]. Similarly, miR-148b also serves as tumor-suppressing miRNA in HCC via targeting the WNT1 Wnt family member 1 protein that is involved in tumor growth [55], while miR-193a-5p significantly inhibits HCC formation and progression, via targeting the nucleolar- and spindle-associated proteins [56], which are involved in genome stability regulation and cell multiplication [57].

Additionally, miR-30e-3p influences the HCC phenotype via targeting mdm2 and activating p53 [58,59], while miR-1249 upregulation is closely associated with the upregulation of the Hedgehog pathway via suppressing PTCH1 expression [60].

Finally, miR-122 is also considered a tumor suppressor for HCC, while it is quite plentiful liver-specific miRNA that has a pivotal role in lipid metabolism [61]. This miRNA is commonly downregulated in HCC tissues, whereas it has been reported that the knockdown of miR-122 leads to steatohepatitis and hepatocarcinogenesis [62]. The aforementioned phenomenon is mainly attributed to the indirect activation of p53 via targeting its regulator, the so-called mouse double minute 2 homolog (mdm2) [63]. In Table 2, we outline a list of tumor-suppressive miRNAs in HCC.

### 3.3. The Interplay between MiRNAs and the Major HCC Predisposing Diseases

Viral hepatitis C and B constitute major risk factors that eventually lead to HCC development [64]. The deregulation of several cell functions, such as cell cycle, signaling, and metabolic pathways, as well as immunity, is mainly attributed to the interplay between miRNA expression levels and the presence of the virus [65]. More particularly, it is reported that HCV replication is modulated by several miRNAs such as miR-141 that amplify HCV replication by reducing the expression of deleted liver cancer 1 (DLC-1), which is a tumor suppressor gene [66]. Another miRNA that alters the anti-viral immunity and enhances HCV replication is miR-122, which constitutes a significant druggable target [67]. An example of the manipulation of miR-122 as a therapeutic target is the application of Miravirsen, which is an anti-miR agent in patients with HCV-genotype 1 (phase2 trial, NCT01200420) [68]. However, there are other miRNAs that have an anti-HCV effect, such as miR-199a, miR-27a, miR-196, as well as let-7b, miR-431, miR-27a, miR-29, and miR-448 [69]. Medical treatment with Interferon-B (INF)-B utilizes the anti-viral effect of the aforementioned miRNAs and increases their expression levels, leading to the suppression of viral replication [70]. Additionally, the expression levels of several genes are either upregulated or suppressed by a wide variety of miRNAs such as CREB1, stearoyl-CoA desaturase (SCD), and PPARG, respectively. Moreover, HBV also induces the deregulation of miRNA expression levels, which eventually leads to the development of HCC [71].

More particularly, miR-122 is not only implicated in HCV but also in HBV replication, presenting high expression levels [72], whereas miR-155, that is upregulated by HBV X protein (HBx), suppresses the replication of HBV via the blockade of the CCAAT/enhancer-binding protein [73], as well as via the activation of the IFN gene expression and the IFN signaling pathway, which enhances cellular resistance against HBV virus [74,75]. Meanwhile, HBV transcription is upregulated via miR-34 downregulation, which is attributed to HBx protein [76].

In addition, miR-122 is also closely implicated in the progression of either alcoholic or non-alcoholic fatty liver diseases (ALD or NAFLD). In both cases, miR-122 plays a key role in the progression of the diseases via altering the lipid metabolic pathway and inducing inflammation [77,78], while there are several miR-122-targeted genes that are closely associated with fatty acid (FA), cholesterol, and triglyceride metabolism, such as diacylglycerol O-acyltransferase 1 (Dgat1) and 1-acyl-sn-glycerol-3-phosphate acyltransferase alpha (Agpat1) genes, which are implicated in endogenous triglyceride synthesis [79]. Meanwhile, it has been demonstrated that the loss of miR-122 in animal models (mice) induces the brisk development of inflammation via the over-secretion of monocyte chemoattractant protein-1 and pro-inflammatory cytokines (TNF-a, IL6) [80], whereas the downregulation of miR-122 is also related with hepatocarcinogenesis and fibrosis progression in non-alcoholic steatohepatitis (NASH) [81].

Moreover, miR-21 downregulation is associated with the progression of the disease from NASH to HCC, as was demonstrated in animal and human HCC models [82]. The aforementioned phenomenon is attributed to the disruption of several signaling pathways, such as STAT3, AKT/PKB, and TGF-b [83].

Furthermore, the suppression and loss of let-7 expression, which is related to alcohol abuse, promotes ALD progression and eventually carcinogenesis, due to the disruption of Lin-28 homolog B (LIN28B) gene expression [84].

Similarly, there are several other miRNAs that are closely implicated in hepatocarcinogenesis and the fibrotic injury of the hepatic parenchyma, such as miR-192, miR-16, miR-199a, miR-15, and miR-497 [85].

Meanwhile, fibrosis is also closely associated with the action of several miRNAs such as miR200, human miR-29, and 34 families, miR-199, that are responsible for the modification of extracellular matrix (ECM), as well as with the activation of hepatic stellate cells (HSCs) [86,87]. It is worth mentioning in the case of the inflamed hepatocytes in NAFLD that there is an increased production of extracellular vesicles that contain several cargoes, such as miR-128-3p and MiR-192, which are closely implicated in the development and progression of the fibrotic injury via interacting with the HSCs, which further produce several fibrogenic proteins [88].

### 3.4. MiRNAs Signature in Premalignant State, HCC Staging, and Prognosis

As was aforementioned, miR-122 normally constitutes a suppressor of hepatic inflammation, while its loss leads to inflammation, fibrosis, and the formation of HCC. However, it is reported that miR-122 expression levels are deregulated not only in malignant but also in premalignant lesions, a phenomenon that usually has a background of chronic viral hepatitis and cirrhosis [86]. These premalignant lesions are characterized by nodularity, and they are subcategorized based on their histological and cytological appearance into (i) macroscopic dysplastic nodules (DNs) and (ii) microscopic dysplastic foci (DF). The former category of DNs is further subclassified based on the grade of atypia into high-grade and low-grade dysplastic nodules (HGDNs, LGDNs). The LGDNs are characterized by reduced expression levels of miR-199b, as well as miR-145, during their advance into small (≤2 cm) HCC, whereas HGDNs that are characterized by a higher tendency of malignant transformation present middling levels of miR-224 compared to LGDNs and HCCs (≤2 cm) that do not present and firmly express, respectively [87].

A wide amount of HCC cases (25%) are attributed to viral hepatitis C; for this reason, there are several bioinformatics analyses that intend to determine non-invasive biomarkers, such as miRNAs for identifying which HCV patients will eventually develop HCC. Some of the circulatory miRNAs that are increased in patients with viral hepatitis C and HCC include miR-3607, miR-215, miR-142, miR-199a, miR-150, miR-224, miR-183, miR-150, miR-424, and miR-217 compared to healthy donors. Meanwhile, there is a notable increase in miR-217 and miR-183 and a decrease in miR-3607 and miR-142 in bioptic HCC specimens [88].

Furthermore, there are several metastasis-related miRNAs demonstrated in animal models such as miR-331-3p, miR-487a, miR-29a, miR-1247-3p, miR-425p, and miR-219-4p that promote metastatic dissemination, and they are associated with a worrisome prognosis [89].

Last but not least, tumor-suppressing miR-199-5p, which is significantly downregulated in HCC, is closely associated with a poor prognosis, large-sized HCC [90], as well as with intravascular tumor thrombus and more advanced tumor–node–metastasis TNM stage [91].

### 3.5. MiRNAs as Predictive and Prognostic Biomarkers for HCC

It is widely demonstrated that HCC prognosis still remains unfavorable, a phenomenon that is mostly attributed to the late diagnostic time, when the disease is already disseminated. There are several conventional HCC biomarkers which are currently used in screening, although their value is limited in clinical practice [92]. The utilization of miRNAs as predictive and prognostic biomarkers in HCC has been in the spotlight in recent years. More particularly, some of the miRNAs that are closely related to a poor prognosis for HCC patients are the following: miR-1468↑, miR-32-5p↑, miR-940↓, miR-221↑, miR-137↓ and miR-296-5p↓, as well as miR-92a↑, miR-638↓, miR-122↓, miR-487a↑, and miR-148↑, with the majority of the aforementioned being associated with viral hepatitis B and C [93,94].

The expression levels of miR-32-5p are found up-regulated and closely associated with multi drug-resistant HCC tumors and the downregulation of PTEN [95]. The low expression of phosphatase and tensin homolog (PTEN) is attributed to the activation of the PI3K/AKT signaling pathway, a phenomenon that leads to tumor progress, drug resistance, epithelial–mesenchymal transition (EMT), as well as neoangiogenesis [95,96]. Meanwhile, upregulated miR-221 targets the PHD finger protein 2 (PHF2) gene, a phenomenon that induces its downregulation, as well as tumor dissemination and worrisome survival rates in HCC patients [97].

Moreover, miR-940 downregulated levels are closely associated with poor overall survival; however, when its expression levels are increased, there is increased apoptosis and reduced growth of the malignant hepatocytes [98]. Similarly, miR-137, miR-296, and miR-638 are also downregulated in HCC cases. More specifically, the former constitutes a significant prognostic biomarker for HCC, being closely implicated with the HCC microenvironment, while it targets the afamin (AFM) gene, leading to tumor invasive behavior and metastatic dissemination [99]. MiR-296-5p is found significantly reduced in HCC tissues compared with the normal tissues in the tumor vicinity, with its normal/increased levels are related to a favorable prognosis, the suppression of EMT phenomenon [50] via downregulating the Nrg1/Erbb signaling pathway, and it exerts a suppressive effect on the stem cell potential of HCC cell lines [50].

Meanwhile, the downregulation of miR-296 is also closely associated with an unfavorable prognosis in HCC patients, whereas its upregulation induces tumor growth suppression and apoptosis, as well as cell cycle downregulation, via targeting the fibroblast growth factor receptor 1 (FGFR1), which plays a key role in cell cycle progression and tumor proliferation [100].

Furthermore, miR-638 is also found to be reduced in the serum of HCC patients, being a novel prognostic factor for HCC [101]. Patients who underwent hepatectomy and had increased levels of miR-638 presented a more frequent development of distant or local metastasis [102]. Additionally, attenuation of its expression levels leads to local invasion and EMT via interacting with the sex determining region Y -box 2 (SOX2) [103]. Nevertheless, it is also related to a favorable prognosis and tumor suppression via inhibiting cyclin D1 [102,104].

Additionally, miR-4258, miR-638, miR-3648, and miR-663a are noticeably up-regulated in HUVECs under the exposure to HuH-7M-derived exosomes, which are closely related to increased vascular permeability and reduced ZO-1 and VE-cadherin expression levels in pre-metastatic stages, while they are secreted by highly aggressive and invasive HCC cell lines [102].

Further, the levels of miR-1246, miR-92a, miR-122, miR-487a, and miR-148a are also upregulated in HCC patients. MiR-1246 and miR-497 are closely associated with tumor staging, diagnosis, and prognosis, while miR-1246 via targeting the activator of the transcription and developmental regulator (AUTS2) gene leads to reduced survival in HCC [105]. MiR-92a is implicated in HCC progression and invasive behavior by interacting with FOXA2 and repressing it [106].

Finally, miR 487a is implicated in HBV-related HCC cases by promoting tumor growth [107]. Meanwhile, miR-148 constitutes a significant biomarker not only for predicting the HCC reoccurrence, but also in HCC screening, especially in cases in which the broadly used alpha fetoprotein (AFP) is low or normal [108], whereas microRNA-148a-3p suppresses HCC progression by downregulating SMAD2 mRNA via integrating with Ago2 [47,109,110].

Lastly, the absence of miR-122 is closely associated with viral hepatitis C infection-related HCC and drug resistance via interacting with several genes and signaling pathways, such as zinc finger protein (Snail)1/2, BCL9, CREB1, and the Wnt signaling pathway, respectively [111].

### 3.6. MiRNAs as Diagnostic or Monitoring Tools

It is demonstrated that salivary miRNAs could be potentially utilized as diagnostic biomarkers in HCC as they are considered highly specific and sensitive [112]. In addition, there are systematic analyses of several miRNAs (>700 miRNAs), which demonstrate a constant trend of overexpressed miRNAs, such as miR-148a-3p, miR-122-5p, miR-885-5p, and miR-125b-5p, as well as miR-100-5p, miR-1974, and miR-365a-3p [113].

Furthermore, miR-34a-5p is considered a promising biomarker for the correlation of HCC with cirrhosis [113], while it has to be underlined that based on bioinformatics analyses, there is an ethnical disparity between Asia, America, and Europe, with the latter having 5–37% upregulated miRNA expression levels. Finally, miR-199a, miR-182-5p, miR-422a, and miR-1269a have been demonstrated as the most auspicious diagnostic biomarkers for HCC [114].

Moreover, it is reported that miR-199a-5p could be potentially used as a monitoring marker as it is closely associated with glucose uptake by cancer cells. The results of miR-199a-5p replacement can be accessed via [18F]-FDG PET-CT performance, which demonstrates the active uptake of glucose by malignant tumor cells [115].

However, there is another promising HCC diagnostic, prognostic, and predictive tool, such as EVs, which are in the spotlight of current research and will be further discussed in Section 4. In Table 3, we demonstrate a summary of the functional role of the aforementioned miRNAs in HCC management either as prognostic, predictive, or diagnostic biomarkers.

## 4. The Interplay between EVs and MiRNAs in HCC

EVs constitute nanostructures that are composed of a bilayered lipid membrane, enclosing a high variety of nucleic acids including miRNAs, long non-coding RNAs (lncRNAs), autophagosomes, mitochondrial DNA, protein, and lipid molecules. They are characterized by great heterogeneity, which is reflected by the fact that they can have a wide variety of sizes, cargoes, and origins [117]. They are subclassified into exosomes, microvesicles, and apoptotic bodies based on their diameter, which is 40–150 nm, 150 nm–1000 nm, and over 1000 nm, respectively. EVs play a pivotal role in cross-talk intercellular communication between the parental cell and the recipients, a phenomenon that is mediated via the uptake of EVs cargoes [118].

Focusing on their functional role in HCC, it is reported that there is a great production of small extracellular vesicles (sEVs) (30–130 nm) from malignant hepatocytes, allowing cross-talk communication between HCC cells, in order to enhance cell proliferation, migration, and invasion, as well as metastatic dissemination [119]. There are several exosomal miRNAs that can be potentially isolated from serum, plasma, and urine, which can be utilized as diagnostic tools and prognostic or predictive biomarkers for HCC [120,121,122,123,124].

In Table 4 and Table 5, we demonstrate some examples of extracellular vesicle (EV)-contained miRNA cargoes, their biological role, and their utilization in HCC management.

## 5. The Interplay between MiRNAs and Autophagy in HCC

Autophagy constitutes a strictly orchestrated homeostatic pathway that reassures the ideal conditions for cell survival under stress, such as under conditions of inadequate nutrients and oxygen, as well as under the augmentation of non-functioning cytoplasmic organelles, which are eventually degraded [139,148]. There are five distinct phases on the autophagy pathway, including the (i) initiation step, (ii) the nucleation of the phagophore, (iii) the phagophore elongation step, (iv) the formation of autophagolysosome, and the (v) cargo degradation step [149,150]. MiRNAs are closely implicated in autophagy regulation, orchestrating each phase of the pathway from the initiation step to the last step of cargo degradation [151]. The emerging role of miRNAs in autophagy orchestration is in the spotlight of studies as produces opens new opportunities for the retrieval of new therapeutic modalities and strategies for the optimal management of HCC. More particularly, several stimuli that induce cellular stress initiate the autophagy pathway, but also some miRNAs that are closely related to autophagy regulation, so-called autophagomiRs [152]. The initiation step is started via the inhibition of the mammalian target of rapamycin (mTOR) and the Unc-51-like kinase1 complex (ULK1) activation, with the former being targeted by several miRNAs such as miR-199a, miR-7, miR-144, miR-7, as well as miR-100 and miR-338-3p [153]. After the engulfment of the cargo, the autophagophore is nucleated, which is achieved via the phosphorylation of class III PI3K by ULK1, followed by Beclin1-PI3K complex formation [154]. Several miRNAs target the components of this step, including the ULK1 suppressive miRNAs such as miR-17-5p, miR-26a-5p, miR-106b, and miR-372. Meanwhile, BECN1 is targeted by miR-181, miR-17/17-5p, and miR-376a/b, which also suppresses the pathway [151,152,155,156], while this is also targeted by activating miRNAs such as miR-221, and it is targeted under chemotherapy and a lack of nutrients by the miR-30 family, miR-9, miR-409-3p, as well as miR-376, miR-199-5p, and miR-20a, respectively [151,152,157,158].

Moreover, Beclin1-PI3K is targeted by activating miR-29b and other suppressive miRNAs such as miR-519a, miR-181a, miR-125a, and miR-374a [151,152,159]. Meanwhile, the elongation of phagophore, including the formation of the autophagosome, is regulated by activating miR-21-3p, miR-9a-5p, miR-155, and miR-20 [151,152,160], while miR-374, miR-519a, miR-520, miR-23b-3p, miR-204, miR-181a, miR-7, and miR-142-3p suppress the elongation step [151,152,160,161].

Focusing on HCC, there are several reports that demonstrate the interplay of autophagy and miRNAs [162]. As it was previously referred to, there are several autophagy-related miRNAs that are either downregulated and upregulated in HCC, such as miR-7 [163], miR-559 [164], miR-101 [165], miR-142-3p [166], miR-181a [167], miR-519d [168], and miR-25 [138], respectively. The deregulation of the expression levels of autophagomiRs is closely related to tumor progression, as well as drug resistance [169].

More particularly interaction between ATG14 (member of class III-PI3K complex) and miR-375 is related to drug resistance (targeted or chemotherapy), including sorafenib, while it is demonstrated that the action of miR-375 on ATG14 leads to the sensitization of HCC cells to sorafenib, via suppressing the cytoprotective effect of autophagy for cancer cells [170].

Similarly, it is also demonstrated that miR-23b-3p modulates the autophagy-induced drug resistance of sorafenib in the HCC HepG2 cell line, while it not only targets ATG12 but also GLS1 (glutaminase), which is related to the high exogenous glutamine that is associated with progressed disease [171].

Moreover, sorafenib resistance in HCC has been also related to miR-25 upregulation, which is closely associated with advanced-stage, lymphatic dissemination and autophagy induction via targeting FBXW7 protein [138,172].

In addition, HCC resistance to sorafenib has been also related to autophagy-related miR-423-5p, which could be potentially utilized as a predictive biomarker for HCC response to sorafenib [173].

Furthermore, the expression levels of miR-519d have been identified as notably increased in HCC, while it induces apoptosis and the autophagy pathway of HCC cells via activating AMPK signaling by binding in the 3ʹ-untranslated region (UTR) of the Ras-related protein 10 (Rab10). More particularly, the overexpressed miR-519d binds to Rab10 3′-UTR, suppressing its expression and the activation (phosphorylation) of AMPK and mTOR [168].

Moreover, it is demonstrated that miR-513b-5p targets the PIK3R3 gene and alters its expression during tumor proliferation and progression in HCC, constituting a potential druggable target [174].

Another miRNA that is downregulated in HCC is miR-559, which normally targets the PARD3 gene. It is shown that the expression level of the aforementioned gene is increased in HCC, a phenomenon that is closely associated with tumor proliferation and progression. Studies have demonstrated that the enhancement of miR-559 expression or the silencing (knockdown) of PARD3 limits the proliferation of HCC cells via autophagy inhibition, as well as inhibits neoangiogenesis via decreasing angiopoietin 2 and vascular endothelial growth factor (VEGF) expression levels [164].

Furthermore, it is demonstrated in hepatoma xenografts (in vivo) that miR-375 suppresses hypoxia-induced autophagy and resensitizes malignant cells to hypoxia, resulting in tumor suppression [175]. Additionally, autophagy-related miR-7 is found to be downregulated in HCC, while it constitutes a tumor suppressor for several malignancies, including HCC. However, the enhancement of miR-7 expression levels limits tumor proliferation via binding on mTOR, while the suppression of autophagy can potentially intensify the anti-proliferative activity of miR-7 in HCC cells [163].

Last but not least, it has to be underlined that autophagy also regulates miRNA expression levels via the selective degradation of several miRNAs, such as miR-224, which constitutes an oncogenic miRNA. More particularly, HBV-associated HCC cases present decreased autophagy activity, resulting in the aggregation of oncogenic miR-224 [176]. We demonstrate in Table 6 several autophagomiRs that are implicated in HCC.

## 6. The Interplay between MiRNAs and Microbiome in HCC

The significance of the human microbiome has been relatively underestimated in recent years; however, there are various studies have been published recently regarding the gut microbiome and its implication in HCC. It is demonstrated that hepatic and gut functions are closely interacting via the gut–liver axis, through the portal circulation. Under microbial dysbiosis, the multilayered gut barrier is disrupted and permeable, leading to the leakage of microbial products, the so-called microbiota-associated molecular patterns (MAMPs) that can interact with liver parenchyma and several hepatic functions, such as the production of bile acids (BAs) [177]. The modification of BAs synthesis is closely related to hepatobiliary malignancy via the deregulation of several cell functions [178,179,180].

More particularly, the baseline gut microbiome is comprised of *Firmicutes*, *Verrucomicrobia*, and *Actinobacteria* phyla, whereas under dysbiosis, there is an overgrowth of *Proteobacteria* and *Bacteroidetes* phyla [181]. It has also been demonstrated that the gut microbiome presents increased levels of lipopolysaccharides (LPS)–Gram-negative bacteria in HCC [182]. HCC miRNA signature includes several miRNAs that are closely associated with hepatic pathophysiological mechanisms, such as miR-21 and miR-666, with the former being positively and negatively correlated with Bacteroides acidifaciens and Firmicutes, respectively, while the latter is also negatively correlated with Firmicutes [183].

The interplay between miRNAs and microbiota is in the spotlight of several types of research for the identification of novel therapeutic targets, as well as diagnostic and prognostic markers. Meanwhile, several tumor-suppressive miRNAs have been demonstrated in female animal models, such as miR-122 and miR-26a/a-1. The expression of the aforementioned tumor suppressive miRNAs has been closely related to farnesoid X receptor (FXR) overexpression, which constitutes a BA receptor, implying gender disparity in HCC risk [183,184].

## 7. Future Therapeutic Opportunities

The utilization of miRNAs and their targets has entered the spotlight of recent research for the development of novel therapeutic strategies. Additionally, new strategies have been demonstrated by taking advantage of the interplay between miRNA and autophagy, as well as the gut microbiome [185]. MiRNAs play a binary role in hepatic carcinogenesis, and their expression levels can be either inhibited or enhanced for achieving tumor suppression [186].

An example of utilizing miRNA targets is the attempt of enhancing the tumor-suppressing effect of miR-122, which targets ADAM10, IGF1R, as well as cyclin G1 and ADAM17. Meanwhile, the suppression of the aforementioned targets could also enhance the expression levels of miR-122 [62].

Additionally, the downregulated levels of miR-296 are notably related to a worrisome prognosis in HCC [50], while the enhancement of its expression levels could limit tumor growth, induce apoptosis, and regulates the cell cycle via interacting with the fibroblast growth factor receptor 1 (FGFR1) gene [100].

Moreover, tumor-suppressing miR-199, which constitutes one of the most downregulated in HCC, can be potentially utilized as a therapeutic target [187]. More particularly, miR-199a-5p is closely related to the regulation of glycometabolism, while under increased levels of expression, it significantly reduces glycolysis products adenosine triphosphate (ATP) and G6P, resulting in the limitation of HCC cells via lowering glucose uptake by malignant hepatocytes [187]. Additionally, another therapeutic modality that is proposed is the utilization of miR-122 injections during trans-arterial chemo-embolization (TACE), which is based on the increased tendency of hepatocytes to uptake short RNA sequences [188].

Furthermore, the silencing of miR-92a could significantly suppress tumor invasion and growth [29], while the enhancement of miR-29a expression levels could also put a break on metastatic dissemination via targeting VEGFA, LOXL2, and LOX [52]. Similarly, the inhibition of oncogenic miR-203a-3p.1, which targets IL24, could also impede HCC metastasis [30].

Moreover, the enhancement of miR-940 expression levels could induce apoptosis and suppress HCC growth, while the same effect was also demonstrated after the inhibition of the estrogen-related receptor gamma (ESRRG), an miR-940 target [98].

Meanwhile, taking advantage of autophagomiRs and their targets on the autophagy pathway, there are several novel therapeutic strategies that can optimize HCC management. More particularly, HCC resistance to sorafenib can be overpassed via targeting miR-25, which is upregulated in HCC via targeting the FBXW7 protein [138,172], while the enhancement of miR-375 action can potentially sensitize the malignant hepatocytes to sorafenib via inhibiting the cytoprotective autophagy pathway in malignant cells [170]. Likewise, the enhancement of miR-559 expression levels could potentially limit HCC growth and drug resistance, as well as neoangiogenesis, via suppressing the expression of PARD3 [164].

In addition, there are miRNAs mimics and inhibitors, such as the miR-221 inhibitor and miR-122 mimic, that demonstrated favorable effects as they significantly reduce proliferation and pro-inflammatory markers, as well as neoangiogenesis [189]. Meanwhile, there is a phase I clinical trial (NCT01829971) for MRX34 which constitutes an miR-34a mimic that is evaluated in advanced solid tumors, including HCC [190]. It was reported that oral dexamethasone as a pre-treatment and daily intravenous MRX34 (daily administration for 5 days with 2 weeks off (21 days in total)) had severe side effects, even resulting in death, thus it was terminated, while the phase II study of MRX34 included a 70 mg/m^2^ dose for HCC [190,191].

Meanwhile, the utilization of gold nanomaterials as miRNA therapeutic agents can potentially restore the miRNA biological function and drug resistance, as they can release miRNA inhibitors or mimics, as well as nanoconjugates, meaning they can carry therapeutic agents that can stimulate endogenous miRNAs [192]. An example of this therapeutic modulation in HCC is the utilization of gold nanoparticles (AuNPs) that transfer miR-326 mimic, which promote its overexpression and the inhibition of the PDK1/AKT/c-myc axis, leading to invasion–migration inhibition, increased apoptosis, and EMT modification [193].

Last but not least, the off-label utilization of amiodarone, which constitutes a widely used antiarrhythmic treatment, could be potentially effective as a suppressor agent for HCC, via inducing autophagy-induced miR-224 degradation [194].

## 8. Conclusions

Hepatocellular carcinoma constitutes a highly aggressive and deadly form of primary liver cancer. Despite all the novel therapeutic modalities, the survival rate for this malignancy remains low. Shedding light on miRNA profiling and its interaction with autophagy, as well as gut microbiome, could potentially open new therapeutic horizons for the development of novel druggable targets. Non-coding miRNAs constitute a strong weapon for the diagnosis and treatment and potentially the monitoring and staging of HCC. Finally, the manipulation of the miRNA signature of HCC tumors could potentially overpass drug resistance, and it can be utilized for the identification and treatment of metastatic HCC tumors. However, further research is considered crucial for the discovery of novel diagnostic and therapeutic perspectives.

## Figures and Tables

**Figure 1 ijms-24-07168-f001:**
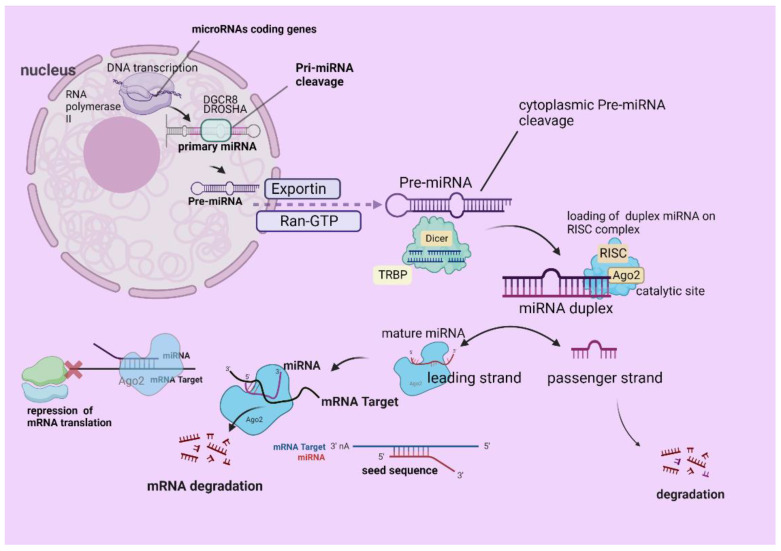
A schematic presentation of miRNA biogenesis. The mechanism of miRNAs biogenesis starts with DNA transcription of specific miRNAs coding genes, under the enzymatic action of RNA polymerase II inside the nucleus. Pri-miRNA arises and is cleaved under the action of DGCR8–Drosha (ribonuclease III), giving rise to pre-miRNA, which is relocated to the cytoplasm via Exportin 5 and Ran-GTP. Pre-miRNA then is cleaved by the Dicer-TRBP, forming the duplex miRNA, which is further loaded on the RISC complex. The active miRNA strand (leading strand/mature miRNA) is loaded on RISC (miRISC) and the other is the so-called passenger, which becomes degraded. The miRISC (seed sequence) binds a target mRNA, which leads to the repression of the mRNA translation, as well as its silencing or degradation. The figure was created with “BioRender.com” (agreement number OS252V998Q).

**Table 1 ijms-24-07168-t001:** A list of oncomiRs, their expression levels, as well as their targets and functional effects in HCC.

OncomiRs	Expression Levels in HCC	Effects	Target
miR-18a	↑	Enhances HCC cell proliferation and tumor migratory behavior	Bcl2l10 [34]
miR-21	↑	Tumor proliferation, migration, and EMT	PI3K/AKT, STAT3, TGF-β/SMADs HEPN1 [32,33]
		Involvement in cell cycle	
miR-92a	↑	Tumor proliferation/invasion	FOXA2 [29]
miR-155-5p	↑	HCC progression	PTEN [35]
miR-203a-3p.1	↑	Metastatic dissemination	IL24 [30]
miR-302d	↑	Tumor proliferation	TGFBR2 [38]
miR-331-3p	↑	HCC progression	E2F1 [36]
miR-346	↑	Tumor growth and progression	BRMS1 [41]
miR-454	↑	Recurrent HCC, metastatic dissemination resistance to therapeutic modalities	TICs [40]
miR-487a	↑	Tumor migration, metastatic dissemination	SPRED2 PIK3R1 [39]
miR-765	↑	Tumor proliferation	INPP4B [37]
miR-873	↑	HCC growth and metastatic dissemination	AKT/mTOR [43]
miR-3910	↑	Tumor progression and migration	MST1 [42]
			Hippo-YAP pathway
miR-4417	↑	Inhibits apoptosis and tumor proliferation	PKM2, TRIM35 [31]

Up-regulated (↑): increased expression levels.

**Table 2 ijms-24-07168-t002:** A list of tumor-suppressive miRNAs in HCC and their effects and targets.

Tumor-Suppressing miRNA	Expression Levels in HCC	Effect	Target
miR-29a	↓	Suppresses invasion	VEGFA, LOXL2 LOX [52,53]
		and metastasis	
miR-30e-3p	↓	Tumor suppression	p53 [58,59]
		Activates p53	
miR-122	↓ or	Regulate lipid metabolism	mdm2, p53 [61,62,63]
	knockdown	Tumor suppression	
miR-148b	↓	Tumor growth suppression	WNT1 [55]
miR-193a-5p	↓	Inhibits HCC formation and progression	Nucleolar- and
			spindle-associated protein [56,57]
miR-195	↓	Suppresses HCC development via interacting the malignant hepatocyte cell cycle.	CDC25A, CDK6, CDK4, CCNE1 [48]
miR-199a	↓	Impairment of tumor growth, proliferation	Aerobic
			Glycolysis [45]
miR-199a-5p	↓	Glycometabolism in hepatocytes	3′ UTR HK2 mRNA [44]
		Reprograms the glycolysis of the malignant cells	
miR-199a/b-3p	↓	Tumor inhibition	MAPK/ERK PAK4 [46]
miR-206	↓	Inhibits HCC development in mice	CDK6, CCND1, cMET [51]
miR-296-5p	↓	Suppresses the HCC stem cell lines/EMT phenomenon	Neuregulin-ERBB pathway [49,50]
miR-766-3p	↓	Tumor suppression	Wnt/β-catenin signaling pathway [54]
miR-1249	↑	Tumor suppression	Hedgehog pathway
			PTCH1 [60]

↑ increased/↓ decreased.

**Table 3 ijms-24-07168-t003:** Summary of miRNAs expression levels and their utilization in HCC management.

Utilization	MiRNAs	Expression Levels	Isolation	References
Diagnostic markers	mir-92b	Decreased in saliva/ increased in tumor tissue	Saliva and Tumor tissue	[112]
	mir-548i-2	Decreased in saliva/ decreased in tumor tissue		
	mir-548l	Decreased in saliva/ decreased in tumor tissue		
Diagnostic markers	miR-1972	Increased only in HCC	Plasma	[113]
	miR-193a-5p	Increased only in HCC		
	miR-214-3p	Increased only in HCC		
	miR-365a-3p	Increased only in HCC		
Poor prognostic markers	miR-137	Decreased	Tumor tissue	[99]
	miR-296	Decreased	Tumor tissue	[100]
	miR-638	Decreased	Serum	[101]
	miR-940	Decreased	Tumor tissue	[98]
	miR-32-5p	Increased	Tumor tissue	[95]
	miR-92a	Increased	Tumor tissue	[106]
	miR-221	Increased	Tumor tissue	[97]
	miR-1246	Increased	Serum	[105]
Good prognostic markers	miR-296-5p	Notably decreased	Tumor tissue	[49,50]
Monitoring markers	miR-199a-5p		[18F]-FDG PET-CT performance	[115]
Follow up for reoccurrence	miR-148		Serum	[108]
Screening markers	miR-148a in ↓ or absent AFP	Increased	Serum	[108]
Staging	Pre-metastatic	Increased	Serum	[102]
	miR-638			
	miR-663			
	miR-3648			
	miR-4258			
Circulatory miRNAs HCV and HCC	miR-142 miR-150	Decreased Decreased	Blood	[116]
	miR-183	Increased		
	miR-199b	Decreased		
	miR-215	Decreased		
	miR-217	Increased		
	miR-224	Increased		
	miR-424	Decreased		
	miR-3607	Decreased		
Bioptic specimens	miR-142		Tumor tissue	[116]
	miR-183			
	miR-217			
	miR-3607			

**Table 4 ijms-24-07168-t004:** EV-contained miRNA cargoes and their utilization as diagnostic, prognostic, and predictive biomarkers in HCC management.

Exosomal miRNA	Isolation	Expression Levels	Utilization
miR-21	Serum	↑ in HCC patients	Diagnostic tool [125]
miR-21-5p	Plasma	↑ in HCC patients	Diagnostic tool and monitoring together with serum AFP [126]
miR-92b	Serum	↑levels—HCC recurrence after liver transplantation	Predictive biomarker [127]
miR-92-3p	Plasma	↑ in HCC patients/AFP low levels	Diagnostic tool and monitoring [126]
miR-101	Serum	↓ in HCC patients	Diagnostic tool [128]
miR-125b	Serum	↓ in HCC patients ↓levels—↓ recurrence time ↓overall survival	Diagnostic [128] and prognostic tool [129]
miR-223-3p	Serum	↓↓ levels—non-responders to TACE treatment	Predictive biomarker [130]
miR-665	Serum	↑ levels—HCC patients ↑↑levels—poor prognosis (stage, tumor size, survival)	Diagnostic tool and prognostic biomarker [121]
miR-718	Serum	↓levels—HCC recurrence after liver transplantation	Predictive biomarker [121]
miR-18a, miR-221, miR-222, miR-224	Serum	↑↑levels—HCC patients	Diagnostic tool [131]
miR-16, miR-146, miR-192	Plasma	↑levels—HCC and cirrhosis	Diagnostic tool and prognostic biomarker [132]
miR-122, miR-148a	Serum		Predictive biomarkers [133]
miR-122	Serum	↓↓ levels post-TACE in Cirrhotic-HCC patients	Predictive biomarkers [134]

↑↑significantly increased/↑ increased/↓↓ significantly decreased/↓ decreased.

**Table 5 ijms-24-07168-t005:** EV-contained miRNA cargoes and their functional role in HCC.

EV-Contained miRNA Cargo	Functional Role in HCC
EVs-miR let-7b [126]	↓tumor inflammation by targeting
	interleukin 6
EVs-miR 15a [135]	Inhibits tumor proliferation/migration
EVs-miR-21 [136]	PDK1/AKT pathway activation
	Conversion of hepatic stellate cells to CAFs
	Tumor migration
	Neoangiogenesis
EVs-miR-23a-3p [137]	Uptake by macrophages
	Promoted tumor escape
	phenomenon
	T-cell function suppression
EVs-miR-25 [121,138]	Modification of BAX, BCL2 expression
	(apoptotic markers for sorafenib)
	Drug resistance in sorafenib
EVs-miR-32-5p [139]	Multi-drug resistance
EVs-miR-103 [140]	Uptake by endothelial cells
	Modified integrity of vessels
	Tumor invasion
EVs-miR-122 [141]	Suppressed tumor proliferation
	↑chemosensitivity
EVs-miR-125a [142]	Targets HCC stem cells CD90
	Suppressed tumor proliferation and
	Migration
EVs-miR-221 [143]	CD44 regulation via interaction with
	PI3K-AKT-mTOR
	Tumor proliferation/migration
EVs-miR-320a [123,144]	Inhibition of MAPK pathway
	Suppressed tumor migration and invasion
EVs-miR-429 [145]	↑E2F1 transcription activity
	Tumor proliferation
EVs-miR-1247-3p [146]	Conversion of fibroblasts to CAFs
	Promoted EMT phenomenon
	Metastatic dissemination and proliferation
EVs-miR 3682 3p [147]	Inhibited neoangiogenesis

↑: increased/↓: decreased.

**Table 6 ijms-24-07168-t006:** The role of autophagomiRs in HCC.

AutophagomiRs	Target	Role in HCC
miR-7 [163]	ATG5	Inhibits invasion and metastasis
miR-25 [138]	FBXW7	Autophagy enhancement Sorafenib resistance
miR-23b-3p [171]	ATG12	↓survival and ↑apoptosis of sorafenib-resistant HepG2 cells
miR-101 [165]	ATG4D RAB5A, STMN1	Inhibits autophagy Enhances cisplatin-induced apoptosis
miR-142-3p [166]	ATG5/ATG16L1	Autophagy inhibition Sensitization of HCC cells to sorafenib
miR-181a [167]	ATG5	autophagy inhibition ↑apoptosis in HepG2 cells ↓tumor growth (HepG2 cell xenograft tumor models)
miR-375 [170]	ATG7	Inhibits autophagy ↓HCC survival under hypoxia
miR-423-5p [173]	Autophagy cell cycle	↑autophagy in HCC cells-treated with sorafenib Partially remitted/stable disease 6 months after the treatment initiation (predictive biomarker)
miR-513b-5p [174]	PIK3R3	Inhibits proliferation, migration/invasion, Induces HCC cell apoptosis
miR-519d [168]	Rab10	Autophagy induction ↑HCC cell Apoptosis AMPK signaling pathway activation via Rab10
miR-559 [164]	PARD3	Inhibits proliferation, neoangiogenesis, inhibits autophagy

↑: increased, ↓: decreased.

## Data Availability

Not applicable.

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
