# Peer review of "An Insight into the Arising Role of MicroRNAs in Hepatocellular Carcinoma: Future Diagnostic and Therapeutic Approaches"

_ijms, 2023, doi:10.3390/ijms24087168_

Round 1

Reviewer 1 Report

In their review article Koustas et al., describe the emerging roles of microRNAs in hepatocellular carcinoma with a focus on the potential use of miRNAs as diagnostic tools and therapeutic targets in HCC. The review is overall well-written, but I have a few comments and points that should be addressed.

1) I think the authors should check again and correct a lot of typos in the text. For example:

Line 72: cannonical

Line 85: afterward

Lines 119, 231 (Hedgehog pathway), 238

In line 83: it should be pre-miRNA, not pre-microRNA and also the pre-miRNA should be defined before (precursor miRNA)

In line 137: what the authors mean by overregulation? Over-expression or up-regulation? I think it is better to change this word. 

All these are just a few examples. There are multiple typos through the whole text. I also encourage the authors to ask a native English speaker to read the manuscript and provide suggestions to improve some expressions. 

Another example: the expression "last but not least" is very repetitive. Please use different expressions with similar meaning. 

2) The paragraphs 3.2.1 and 3.2.2 do not provide any useful information except for mentioning a lot of different miRNAs that is impossible for a reader to remember their functions in HCC. I recommend the authors to make three extra tables and categorize these miRNAs according to their regulation in HCC (up- or down-regulation) and their biological function (HCC invasion/migration, metastatic dissemination, drug resistance etc).

3) In section 7 (future therapeutic opportunities) the authors mention that miRNAs are good therapeutic targets in HCC, but they do not describe how they can be targeted. For example miRNA inhibitors and mimics have been generated over the years, but is it feasible to target a specific miRNA without side effects, considering that one miRNA can have multiple target mRNAs? What are the limitations? In addition, are there ongoing or completed clinical trials using miRNA inhibitors or mimics for HCC? 

4) Finally, although very correctly the authors state that miRNAs can be used as diagnostic biomarkers, they did not discuss at all about those miRNAs found within extracellular vesicles (EVs), which can travel away from the tumor sites and can be found and isolated in human fluids, such as, blood, serum, urine etc. What is known about miRNAs enriched in EVs from patients with HCC? It is worth to add an extra section describing this important aspect, as miRNAs within EVs could be ideal biomarkers and they could be identified through non-invasive tests for HCC patients. Also, it would be nice to generate an extra table with a few examples of miRNAs found in EVs in HCC cell lines or even HCC patients, if there is literature about this. 

Author Response

UNIVERSITY  OF  ATHENS  MEDICAL  SCHOOL

March 11, 2023

Internation Journal of Molecular Science

RE: Submission of REVISED ‘Review’ (ijms-2232683)

Dear Editor

Please find enclosed our REVISED Mini-Review entitled An insight into the arising role of microRNAs in Hepatocellular carcinoma: Future diagnostic and therapeutic approaches to be considered for publication. We would like to thank you and the reviewers for your thoughtful evaluation of our manuscript and your most welcome comments/suggestions. Accordingly, we have now revised our manuscript thoroughly to reflect these comments.

Please find below a point-by-point response to ALL the issues raised by the Reviewers:

Reviewer #1:

Comment-1: I think the authors should check again and correct a lot of typos in the text. For example:

  1. Line 72: cannonical
  2. Line 85: afterward
  3. Lines 119, 231 (Hedgehog pathway), 238
  4. In line 83: it should be pre-miRNA, not pre-microRNA and also the pre-miRNA should be defined before (precursor miRNA)
  5. In line 137: what the authors mean by overregulation? Over-expression or up-regulation? I think it is better to change this word.

AUTHOR RESPONSE: Firstly, we would like to thank the Reviewer for his/her constructive and thorough examination of our manuscript that helps us to improve it. The Manuscript is currently revised also at English level. Please see below for detailed responses to his/her minor/major comments.

  1. “Cannonical” has been replaced with “Canonical”
  2. “Afterward” has been replaced with “Afterwards”
  3. “Hedgehog signaling pathway” replaced the typing mistakes
  4. “Pre-miRNA” replaced “pre-microRNA” and we added the definition of pre-miRNA as precursor
  5. “Overregulation” was used as synonym with “upregulated”, however we replaced it by up-regulation as you recommended.

Comment-2: There are multiple typos through the whole text. I also encourage the authors to ask a native English speaker to read the manuscript and provide suggestions to improve some expressions.

AUTHOR RESPONSE: We thank the reviewer. We have made the appropriate editing. The Manuscript is currently revised also at English level.

Comment-3: Another example: the expression "last but not least" is very repetitive. Please use different expressions with similar meaning.

AUTHOR RESPONSE: We thank the reviewer. We have made the appropriate editing. Last but not least has been replaced with synonym words.

Comment-4 :The paragraphs 3.2.1 and 3.2.2 do not provide any useful information except for mentioning a lot of different miRNAs that is impossible for a reader to remember their functions in HCC. I recommend the authors to make three extra tables and categorize these miRNAs according to their regulation in HCC (up- or down-regulation) and their biological function (HCC invasion/migration, metastatic dissemination, drug resistance etc).

AUTHOR RESPONSE: We thank the reviewer. We have made the appropriate editing. We reconstructed the skeleton of the manuscript. We strictly made two tables 1 & 2 with oncomiRs and tumor suppressing miRNAs, respectively including their expression levels in HCC tissue, target and effect on HCC. Moreover, we removed the paragraphs with the simple miRNA reference, due to the fact that we lose the attention of the reader.

Comment-5: In section 7 (future therapeutic opportunities) the authors mention that miRNAs are good therapeutic targets in HCC, but they do not describe how they can be targeted. For example, miRNA inhibitors and mimics have been generated over the years, but is it feasible to target a specific miRNA without side effects, considering that one miRNA can have multiple target mRNAs? What are the limitations? In addition, are there ongoing or completed clinical trials using miRNA inhibitors or mimics for HCC? 

AUTHOR RESPONSE: We thank the reviewer. We added in the section of future therapeutic opportunities (in the revised manuscript some ongoing or completed clinical trials using miRNA inhibitors or mimics for HCC, as you suggested.

Comment-6 : Finally, although very correctly the authors state that miRNAs can be used as diagnostic biomarkers, they did not discuss at all about those miRNAs found within extracellular vesicles (EVs), which can travel away from the tumor sites and can be found and isolated in human fluids, such as, blood, serum, urine etc. What is known about miRNAs enriched in EVs from patients with HCC? It is worth to add an extra section describing this important aspect, as miRNAs within EVs could be ideal biomarkers and they could be identified through non-invasive tests for HCC patients. Also, it would be nice to generate an extra table with a few examples of miRNAs found in EVs in HCC cell lines or even HCC patients, if there is literature about this. 

AUTHOR RESPONSE: We thank the reviewer. We added more information about the utilization of EVs as diagnostic biomarkers in section, as you recommended in the new section 4. The interplay between EVs and miRNAs in HCC and we added Table 4&5., where we demonstrate some examples of EV-contained miRNA cargoes and their role in HCC, including the exosomal-miRNAs that are used as diagnostic biomarkers (serum/plasma).

Trusting that we have adequately addressed the reviewers' concerns, we would like to thank you for your help in improving our work significantly.

Kind regards,

Koustas Evangelos, MD, PhD

Reviewer 2 Report

In the review, Koustas et al. present a comprehensive overview of miRNAs that act as oncomiRs or tumor suppressor miRNAs in hepatocellular carcinoma (HCC). They highlight the role of miRNAs in autophagy in HCC and conclude with short chapters on the potential use of miRNAs in diagnostic and therapeutic applications. Overall, the review is well structured and considers recent findings in the field. However, several references are missing and the manuscript requires some modifications.

Major

1.      What selection criteria were used in the literature search for this review article?

2.      The authors describe miRNAs classified either as tumor suppressors or tumor promoters. The term “tumor promoter” should be replaced by “oncomiR” throughout the manuscript because the loss of a tumor suppressor miRNA also promotes tumor formation. OncomiR or onco-miRNA is the common term.

3.      Chapter 2: The description of the miRNA biogenesis pathway is not precise and would profit from rephrasing.  Note that the mature miRNA is not a duplex, RISC is not cleaving the miRNA strand, and the binding of a miRNA to its target mRNA requires only partial complementarity (keyword: seed sequence). Replace the term “biogenetic mechanism” by “biogenesis” throughout the manuscript.

4.      Table 1: The structure of the table is not self-explaining. The authors should consider another visual representation of the data. Additionally, information about the size of the patient cohorts analyzed in the different cited studies would further add value to the presented data.

5.      Chapter 3.2 and sub-chapters: The chapters cover both, oncomiRs and tumor suppressor miRNAs. Why are down-regulated miRNAs included? In respect to the definition of oncomiRs and tumor suppressor miRNAs this is not reasonable in respect to the title of the chapter. Following sentences of the chapters list down-regulated miRNAs:  lines 202-205, lines 223-237, lines 242-244,  lines 247-249, lines 250-251, lines 253-255. The tumor suppressor miRNAs should be discussed in chapter 3.3, only. The clear discrimination will also avoid repetitions in the text.

6.      Table 2: Should list oncomiRs, only. The tumor suppressor miRNAs should only be listed in table 3 and not in table 2. The miRNAs should be ordered by number in each column.

Alternatively, the authors could present the information from table 2 and 3 in a Venn diagram using a color code to distinguish between oncomiRs and tumor suppressor miRNAs. This would add value by indicating which miRNAs are involved in multiple processes. Regardless of tabular or graphical representation, references must be listed next to each miRNA.

7.      In different chapters miR-148a/b is described as up-regulated (lines 175 -185, 518) and down-regulated miRNA (lines 234-235, 271, 292-294). Similarly, miR-122 is listed as up-regulated (line 175), and down-regulated (lines 186, 297-303) in HCC. The authors do not comment on this conflicting observations and need to discuss the different roles observed.

8.      Chapter 4: The authors describe the cellular phases of autophagy and the miRNAs that interact with the various players of the pathway in HCC. The authors might consider adding a figure summarizing the role of autophagomiRs in HCC or a table listing autophagomiRs with known relevance in HCC.

9.      Chapter 6: This chapter remains too superficial, especially, in respect to the title of the review that claims a focus on diagnostic approaches. The authors need to make clear in which kind of samples (e.g. salivary, blood) the miRNAs are used as biomarkers. Are all examples listed from salivary? An overview about serum miRNAs as biomarkers for HCC is lacking although there are many data in the literature.

10.   Chapter 7: The title of the review indicates a focus on future therapeutic approaches. Therefore, chapter 7 needs improvement. At what stage between bench and bedside are the presented therapeutic options? Are any of the examples in clinical trials?  The chapter would benefit from an assessment or discussion by the authors of the status quo of the therapeutic attempts. Information about the applications of miR-122 (lines 537-540 and lines 550-553) should be provided in one paragraph and not scattered.

Minor

1.      English needs some revision.

2.      There are many typos and missing spaces throughout the manuscript. Especially, correct writing of miRNA names has to be ensured.

3.      The up- or down-regulation of a miRNA in a sample is always observed in comparison to a control. It depends also on the control, if the miRNA is up-, or down-regulated. To be precise, the corresponding controls should be mentioned in the sentences wherever necessary in the manuscript (e.g. lines 136, 141, 175, 518).

4.      Lines 43-44: Reference is missing.

5.      Line 52: Use capital letters for hepatitis B and C.

6.      Line 81: typo, the correct gene name is DGCR8.

7.      Line 86: typo, correct is Ran-GTP.

8.      Figure 1 and legend to Figure 1: What is “GTP6”? The GTPase Ran can bind one GTP molecule. Correct is Ran-GTP. There is also a typo in the microprocessor subunit “DGR8”, correct is DGCR8.

9.      Chapter 3: The authors should avoid a mere stringing together of examples (e.g. lines 198-205). It is usually helpful to refer the reader to tables or visual representations.

10.   Lines 130-132: References are missing.

11.   Line 136: Reference is missing.

12.   Lines 141-143: Reference is missing.

13.   Lines 156-157: Reference is missing.

14.   Lines 165-166: Reference is missing.

15.   Line 184: AFP is the abbreviation for what?

16.   Lines 246-251: References are missing.

17.   Lines 278-280: The sentence needs to be reworded and a reference is missing.

18.   Lines 285-286: Reference is missing.

19.   Lines 297-299: Reference is missing.

20.   Lines 324-325: Reference is missing.

21.   Lines 335-337: Reference is missing.

22.   Lines 337-341: Reference is missing.

23.   Lines 357-360: Reference is missing.

24.   Chapter 3.5: The authors might consider adding a brief description of the typical premalignant states and the progression to HCC at the beginning of the chapter.

25.   Line 374: …not only in malignant but only also in premalignant lesions…

26.   Lines 393-396: Reference is missing.

27.   Lines 422-425: Reference is missing.

28.   Lines 428-430: Reference is missing.

29.   Lines 430-433: Reference is missing.

30.   Lines 442-444: The sentence is not clear and a reference is missing.

31.   Lines 447-448: Reference is missing.

32.   Lines 466-467: Reference is missing.

33.   Lines 490-496: References are missing.

34.   Lines 498-500: Reference is missing.

35.   Lines 509-513: The link between the given example and the HCC microbiome is lacking.

36.   Lines 537-539: Reference is missing.

37.   Lines 554-556: References are missing.

Author Response

UNIVERSITY  OFATHENSMEDICAL  SCHOOL

March11, 2023

Internation Journal of Molecular Science

RE:Submission of REVISED‘Review’ (ijms-2232683)

Dear Editor

Please find enclosed our REVISED Mini-Review entitled An insight into the arising role of microRNAs in Hepatocellular carcinoma: Future diagnostic and therapeutic approachesto be considered for publication. We would like to thank you and the reviewers for your thoughtful evaluation of our manuscript and your most welcome comments/suggestions. Accordingly, we have now revised our manuscript thoroughly to reflect these comments.

Please find below a point-by-point response to ALL the issues raised by the Reviewers:

Reviewer #2:

Comment-1: What selection criteria were used in the literature search for this review article?

AUTHOR RESPONSE: Firstly, we would like to thank the Reviewer for his/her constructive and thorough examination of our manuscript that helps us to improve it. The Manuscript is currently revised also at English level. Please see below for detailed responses to his/her minor/major comments.

Our selection criteria for the literature search are: 

  • reviews in databases such as PubMed or MEDLINE/ trial registries
  • Recent studies 2018-2023
  • Reviews from most respectable journals
  • Search in the database that includes synonyms, related terms and alternative phrases, acronyms, spelling variants.

Comment-2

The authors describe miRNAs classified either as tumor suppressors or tumor promoters. The term “tumor promoter” should be replaced by “oncomiR” throughout the manuscript because the loss of a tumor suppressor miRNA also promotes tumor formation. OncomiR or oncomiRs is the common term.

AUTHOR RESPONSE: We thank the reviewer.We reconstructed the skeleton of this manuscript. We start with oncomiRs, tumor suppressor separately with two more detailed and clarifying Tables.

Comment-3

Chapter 2: The description of the miRNA biogenesis pathway is not precise and would profit from rephrasing.  Note that the mature miRNA is not a duplex, RISC is not cleaving the miRNA strand, and the binding of a miRNA to its target mRNA requires only partial complementarity (keyword: seed sequence). Replace the term “biogenetic mechanism” by “biogenesis” throughout the manuscript.

AUTHOR RESPONSE: We thank the reviewer. We rephrased this section in order to clarify the biogenesis of miRNA, which had some inaccuracies and expressional mistakes. We clarified how the duplex miRNA is formed, the mature miRNA (which is not duplex) and the role of RISC.

We added information about the partial complementarity and the seed sequence as you recommended. we replaced biogenetic mechanism by biogenesis as you recommended.

Comment-4:

Table 1: The structure of the table is not self-explaining. The authors should consider another visual representation of the data. Additionally, information about the size of the patient cohorts analyzed in the different cited studies would further add value to the presented data.

AUTHOR RESPONSE: We thank the reviewer.We reconstructed the skeleton of this manuscript. We start with oncomiRs, tumor suppressor separately with two more detailed and clarifying Tables.

Additionally, this review is already long and full of information, so we did not believe that adding the size of patient cohorts of the cited studies will give further information about the subject of this manuscriptand we may lose the attention of the reader.

Comment-5:

Chapter 3.2 and sub-chapters: The chapters cover both, oncomiRs and tumor suppressor miRNAs. Why are down-regulated miRNAs included? In respect to the definition of oncomiRs and tumor suppressor miRNAs this is not reasonable in respect to the title of the chapter.

Following sentences of the chapters list down-regulated miRNAs:  lines 202-205, lines 223-237, lines 242-244, lines 247-249, lines 250-251, lines 253-255. The tumor suppressor miRNAs should be discussed in chapter 3.3, only. The clear discrimination will also avoid repetitions in the text.

AUTHOR RESPONSE: We thank the reviewer. We strictly separated oncomiRs from tumor suppressor miRNAs. The expression levels of each miRNA are referred. We avoid the repetitions in the revised manuscript. We made clarifying tables

Comment-6:

Table 2: Should list oncomiRs, only. The tumor suppressor miRNAs should only be listed in table 3 and not in table 2. The miRNAs should be ordered by number in each column.

Alternatively, the authors could present the information from table 2 and 3 in a Venn diagram using a color code to distinguish between oncomiRs and tumor suppressor miRNAs. This would add value by indicating which miRNAs are involved in multiple processes. Regardless of tabular or graphical representation, references must be listed next to each miRNA.

AUTHOR RESPONSE: We thank the reviewer.The Tables are reconstructed and they include only oncomiRs or suppressors, their targets, their effect and their expression levels.

Comment-7:

In different chapters miR-148a/b is described as up-regulated (lines 175 -185, 518) and down-regulated miRNA (lines 234-235, 271, 292-294). Similarly, miR-122 is listed as up-regulated (line 175), and down-regulated (lines 186, 297-303) in HCC. The authors do not comment on this conflicting observations and need to discuss the different roles observed.

AUTHOR RESPONSE: We thank the reviewer.

In the revised manuscript in line 199-201 in the section

3.2. miRNAs as tumor suppressors for HCC

“miR-148b also serves as tumor-suppressing miRNA in HCC via targeting WNT1 Wnt family member 1 protein that is involved in tumor growth [55]”

Based on the literature https://www.nature.com/articles/srep08087 the expression levels of miR-148b are downregulated,while overexpression of miR-148b inhibited HCC HepG2 cells proliferation and tumorigenicity.

In the revised manuscript in line 355-360

Section 3.5. MiRNAs as predictive and prognostic biomarkers for HCC

MiR-148a upregulation is seen in hepatitis B associated hepatocellular carcinoma, while utilization of anti-miR-148a suppress cell proliferation, cell cycle progression, cell migration.

https://pubmed.ncbi.nlm.nih.gov/22496917/

https://www.ncbi.nlm.nih.gov/pmc/articles/PMC3322146/#!po=10.0000

lines 312-313 miR-148↑ in HCC patients associated with viral hepatitis B and C

Lines  section3.6 MiRNAs as diagnostic or monitoring tools

MiR-148a-3p are overexpressed

(113)    Jin, Y.; Wong, Y. S.; Goh, B. K. P.; Chan, C. Y.; Cheow, P. C.; Chow, P. K. H.; Lim, T. K. H.; Goh, G. B. B.; Krishnamoorthy, T. L.; Kumar, R.; Ng, T. P.; Chong, S. S.; Tan, H. H.; Chung, A. Y. F.; Ooi, L. L. P. J.; Chang, J. P. E.; Tan, C. K.; Lee, C. G. L. Circulating MicroRNAs as Potential Diagnostic and Prognostic Biomarkers in Hepatocellular Carcinoma. Sci. Rep. 2019, 9 (1), 10464. https://doi.org/10.1038/s41598-019-46872-8.          

We modified the tables as you proposed.

Comment-8:

Chapter 4: The authors might consider adding a figure summarizing the role of autophagomiRs in HCC or a table listing autophagomiRs with known relevance in HCC.

AUTHOR RESPONSE: We thank the reviewer.We added Table 6. with autophagomiRs and HCC, as you recommended.

Comment-9:    Chapter 6: This chapter remains too superficial, especially, in respect to the title of the review that claims a focus on diagnostic approaches. The authors need to make clear in which kind of samples (e.g. salivary, blood) the miRNAs are used as biomarkers. Are all examples listed from salivary? An overview about serum miRNAs as biomarkers for HCC is lacking although there are many data in the literature.

AUTHOR RESPONSE: We thank the reviewer.  We added a distinct section about Extracellular vesicles asdiagnostic tools, predictive and prognostic tools in serum or plasma for HCC identification, as well as a detailed Table 4 & 5.

Comment-10: Chapter 7: The title of the review indicates a focus on future therapeutic approaches. Therefore, chapter 7 needs improvement. At what stage between bench and bedside are the presented therapeutic options? Are any of the examples in clinical trials?  The chapter would benefit from an assessment or discussion by the authors of the status quo of the therapeutic attempts. Information about the applications of miR-122 (lines 537-540 and lines 550-553) should be provided in one paragraph and not scattered

AUTHOR RESPONSE: We thank the reviewer. We added in the section 8 Future therapeutic opportunities some ongoing or completed clinical trials using miRNAs mimics and inhibitors such as miR-221 inhibitor and miR-122 mimic

phase I clinical trial (NCT01829971) for MRX34, which constitutes a miR-34a mimic, as well as about the utilization of gold nanomaterials as miRNA therapeutic agents.

Comments:Minor

  1. English needs some revision.
  2. There are many typos and missing spaces throughout the manuscript. Especially, correct writing of miRNA names has to be ensured.
  3. The up- or down-regulation of a miRNA in a sample is always observed in comparison to a control. It depends also on the control, if the miRNA is up-, or down-regulated. To be precise, the corresponding controls should be mentioned in the sentences wherever necessary in the manuscript (e.g. lines 136, 141, 175, 518).
  4. Lines 43-44: Reference is missing.
  5. Line 52: Use capital letters for hepatitis B and C.
  6. Line 81: typo, the correct gene name is DGCR8.
  7. Line 86: typo, correct is Ran-GTP.
  8. Figure 1 and legend to Figure 1: What is “GTP6”? The GTPase Ran can bind one GTP molecule. Correct is Ran-GTP. There is also a typo in the microprocessor subunit “DGR8”, correct is DGCR8.
  9. Chapter 3: The authors should avoid a mere stringing together of examples (e.g. lines 198-205). It is usually helpful to refer the reader to tables or visual representations.
  10. Lines 130-132: References are missing.Line 136: Reference is missing.
  11. Lines 141-143: Reference is missing.
  12. Lines 156-157: Reference is missing.
  13. Lines 165-166: Reference is missing.
  14. Line 184: AFP is the abbreviation for what?
  15. Lines 246-251: References are missing.
  16. Lines 278-280: The sentence needs to be reworded and a reference is missing.
  17. Lines 285-286: Reference is missing.
  18. Lines 297-299: Reference is missing.

  1. Lines 324-325: Reference is missing.
  2. Lines 335-337: Reference is missing.
  3. Lines 337-341: Reference is missing.
  4. Lines 357-360: Reference is missing.
  5. Chapter 3.5: The authors might consider adding a brief description of the typical premalignant states and the progression to HCC at the beginning of the chapter.
  6. Line 374: …not only in malignant but only also in premalignant lesions…
  7. Lines 393-396: Reference is missing.
  8. Lines 422-425: Reference is missing.
  9. Lines 428-430: Reference is missing.
  10. Lines 430-433: Reference is missing.
  11. Lines 442-444: The sentence is not clear and a reference is missing.
  12. Lines 447-448: Reference is missing.
  13. Lines 466-467: Reference is missing.
  14. Lines 490-496: References are missing.
  15. Lines 498-500: Reference is missing.
  16. Lines 509-513: The link between the given example and the HCC microbiome is lacking.
  17. Lines 537-539: Reference is missing
  18. Lines 554-556: References are missing.

AUTHOR RESPONSE: We thank the reviewer.The Manuscript has been currently revised also at the English level and typos are corrected. Please see below for detailed responses to his/her minor comments.

Capital letters are currently used for HBV and HCV (hepatitis B andC).

The GTP6 typing error was replaced by GTP, as well as DGCR8 replaced the typo.

The figure was modified.

Line 184: AFP is an abbreviation for Alpha Fetoprotein. The explanation is currently added.

We rephrased the lines 278-280 and we added the reference (in the revised manuscript 185-188 lines)

In Chapter 3.5 we did not want to emphasize also in that field, due to the fact that the manuscript is already full of information. However, we already describe the types of premalignant lesions

“These premalignant lesions are characterized by nodularity and they are subcategorized based on their histological and cytological appearance into (i) macroscopic dysplastic nodules (DNs) and (ii) microscopic dysplastic foci (DF).  The former category of DNs is further subclassified based on the grade of atypia into high-grade and low-grade dysplastic nodules (HGDNs, LGDNs)”.

 And we discuss the evolution towards HCC in section 3.4 interplay between miRNA and major predisposing diseases.

The Line 374: …not only in malignant…  was corrected.

Lines 509-513: There is an abrupt passage from one section to another, that’s why we modified the sequence of the sections and we made a smooth transition

The references you underlined are added in the revised manuscript. More particularly:

43-44 : Reference was added (line 43-44)

 130-132: The skeleton of this manuscript was reconstructed. This sentence is no more on the manuscript.

136 : we did not include this sentence in the revised manuscript.

 141-143: The reference was added (lines 315-318) in the revised manuscript

156-157:The reference was added (lines 328-330) in the revised manuscript

165-166: The reference was added ( line 338-339) in the revised manuscript

246-251: We modified the skeleton of this manuscript, so this long sentence is no more on the manuscript.

285-286: The reference was added (lines 193-194) in the revised manuscript

297-299: we added more references as you recommended (lines 202-205)

324-325: The reference was added (lines240-242) in the revised manuscript

335-337:The reference was added (lines 251-255) in the revised manuscript

337-341:The reference was added (lines 254-258) in the revised manuscript

357-360:The reference was added (lines 271-274) in the revised manuscript

393-396: The reference was added more references (lines 300-303) in the revised manuscript

422-425:  The reference were added in the revised manuscript 462-474)

428-430: we did not include this sentence in the revised manuscript

430-433:The reference was added (lines 477-479) in the revised manuscript

442-444: The reference was added (lines 487-490) in the revised manuscript

447-448:The reference was added (lines 491-493)revised manuscript

466-467:The reference was added (lines503-505) in the revised manuscript

490-496:The reference was added (lines 537-547) in the revised manuscript

498-500:The reference was added (lines 544-547) in the revised manuscript

537-539:The reference was added (lines 569-572) in the revised manuscript

554-556: The reference was added (lines 584-586) in the revised manuscript

Trusting that we have adequately addressed the reviewers' concerns, we would like to thank you for your help in improving our work significantly.

Kind regards,

Koustas Evangelos,MD, PhD

Round 2

Reviewer 2 Report

Koustas et al. have considerably improved the manuscript and adequately addressed my concerns. Specifically, the sections about the role of miRNAs for diagnosis and therapy in HCC were expanded. However, a few minor points should still be considered prior publication.

Minor

Tables: in general, miRNAs listed should be ordered by number and homogenously start with miR… (currently it is a mixture of MiR and miR)

Line 34: replace “promoters” by “oncomiRs”

Line 215: replace “tumor promoters” by “oncomiRs”

Line 238: replace “STA3” by “STAT3”

Table 1: Target genes of miR-21, replace “STA3” by “STAT3”

Line 372: replace “miR155” by “miR-155”

Line 407: add the abbreviation “EV”

Lines 434-437: Reference is missing.

Lines 476-480: This sentence is very long and not easy to understand.

Line 496: …while there are being secreted…

Table 3: According to the information given in the text the table contains miRNAs analyzed in different types of samples. It needs to become clear in which type of samples (tumor tissue, serum, salivary, etc ) the listed miRNAs are utilized as marker. It is necessary to add a column for the information about the type of sample in the table (like it is provided in table 4). References appear in both columns, please provide them next to the miRNAs or add an additional column for the references.

Line 636: the sentence seems incomplete. Interaction of ATG14 with ?

Chapter 5: Somewhere in the chapter the authors should refer to table 6.

Line 732-733: Reference is missing.

Line 748-750: Reference is missing.

Author Response

March 23, 2023 Internation Journal of Molecular Science RE: Submission of REVISED ‘Review’ (ijms-2232683) Dear Editor Please find enclosed our REVISED Mini-Review entitled “An insight into the arising role of microRNAs in Hepatocellular carcinoma: Future diagnostic and therapeutic approaches” to be considered for publication. We would like to thank you and the reviewers for your thoughtful evaluation of our manuscript and your most welcome comments/suggestions. Accordingly, we have now revised our manuscript thoroughly to reflect these comments. Please find below a point-by-point response to ALL the issues raised by the Reviewers: Reviewer #2: Comment 1: Tables: in general, miRNAs listed should be ordered by number and homogenously start with miR… (currently it is a mixture of MiR and miR). AUTHOR RESPONSE: Firstly, we would like to thank the Reviewer for his/her constructive and thorough examination of our manuscript that helps us to improve it. We made all the modifications that you suggested. Comment 2: Line 34: replace “promoters” by “oncomiRs” AUTHOR RESPONSE: We thank the reviewer. We replaced promoters by oncomiRs. Comment 3: Line 215: replace “promoters” by “oncomiRs” AUTHOR RESPONSE: We thank the reviewer. We replaced promoters by oncomiRs in the whole manuscript as you recommended. (Such as in line 129 and 134 in the revised manuscript) Comment 4:Line 238: replace “STA3” by “STAT3” AUTHOR RESPONSE: We thank the reviewer. We corrected the typing error Comment 5: Table 1: Target genes of miR-21, replace “STA3” by “STAT3” AUTHOR RESPONSE: We thank the reviewer. We corrected the typing error Comment 6:Line 372: replace “miR155” by “miR-155” AUTHOR RESPONSE: We thank the reviewer. We replaced miR155 by miR-155 Comment 7: Line 407: add the abbreviation “EV” AUTHOR RESPONSE: We thank the reviewer. We added the abbreviation Comment 8: Lines 434-437: Reference is missing. AUTHOR RESPONSE: We thank the reviewer. The manuscript has been significantly reconstructed so in line 434-437 that are part of a Table we have already put references for these lines. Comment 9 :Lines 476-480: This sentence is very long and not easy to understand. AUTHOR RESPONSE: We thank the reviewer. We rephrased this part. Comment 10 Line 496: …while there are being secreted… AUTHOR RESPONSE: We thank the reviewer. In the Revised Manuscript or the initial there is no this sentence in line 496. However, we searched this expression in the whole revised manuscript “while there are being secreted by highly aggressive and invasive HCC cell lines [102].” Lines 349-350. Comment 11 Table 3: According to the information given in the text the table contains miRNAs analyzed in different types of samples. It needs to become clear in which type of samples (tumor tissue, serum, salivary, etc ) the listed miRNAs are utilized as marker. It is necessary to add a column for the information about the type of sample in the table (like it is provided in table 4). References appear in both columns, please provide them next to the miRNAs or add an additional column for the references. AUTHOR RESPONSE: We thank the reviewer. We made the modificationsin Table 3 and 4 as you suggested. Comment 12 Line 636: the sentence seems incomplete. Interaction of ATG14 with? AUTHOR RESPONSE: We thank the reviewer. We don’t have any sentence in line 636 (in the revised or the initial) but the author contributions. However, in line 492 that we referred to ATG14 we have clarified the sentence. “More particularly interaction between ATG14 (member of class III-PI3K complex) and miR-375” Comment 13: Chapter 5: Somewhere in the chapter the authors should refer to table 6. AUTHOR RESPONSE: We thank the reviewer. We added a reference about this table in line 534 Comment 14:Line 732-733: Reference is missing. AUTHOR RESPONSE: We thank the reviewer. These lines do not correspond to any sentence in the revised or the initial manuscript. We noticed that we had a missing reference in lines 581-582 in the revised manuscript , which was added. Comment 15:Line 748-750: Reference is missing. AUTHOR RESPONSE: We thank the reviewer. A it was mentioned in comment 14 answer these lines do not correspond to any sentence in the revised or the initial manuscript. We noticed that we had a missing reference in lines 581-582 in the revised manuscript , which was added. Trusting that we have adequately addressed the reviewers' concerns, we would like to thank you for your help in improving our work significantly. Kind regards, Koustas Evangelos, MD, PhD Corresponding author
